# UPDeT: Universal Multi-agent Reinforcement Learning via Policy Decoupling with Transformers

**Siyi Hu[1], Fengda Zhu[1], Xiaojun Chang[1]\*, Xiaodan Liang[2,3]**
[1]Monash University, [2]Sun Yat-sen University, [3]Dark Matter AI Inc.
{siyi.hu,fengda.zhu}@monash.edu {cxj273,xdliang328}@gmail.com

## Abstract

Recent advances in multi-agent reinforcement learning have been largely limited training one model from scratch for every new task. This limitation occurs due to the restriction of the model architecture related to fixed input and output dimensions, which hinder the experience accumulation and transfer of the learned agent over tasks across diverse levels of difficulty (e.g. 3 vs 3 or 5 vs 6 multi-agent games). In this paper, we make the first attempt to explore a universal multi-agent reinforcement learning pipeline, designing a single architecture to fit tasks with different observation and action configuration requirements. Unlike previous RNN-based models, we utilize a transformer-based model to generate a flexible policy by decoupling the policy distribution from the intertwined input observation, using an importance weight determined with the aid of the self-attention mechanism. Compared to a standard transformer block, the proposed model, which we name Universal Policy Decoupling Transformer (UPDeT), further relaxes the action restriction and makes the multi-agent task's decision process more explainable. UPDeT is general enough to be plugged into any multi-agent reinforcement learning pipeline and equip it with strong generalization abilities that enable multiple tasks to be handled at a time. Extensive experiments on large-scale SMAC multi-agent competitive games demonstrate that the proposed UPDeT-based multi-agent reinforcement learning achieves significant improvements relative to state-of-the-art approaches, demonstrating advantageous transfer capability in terms of both performance and training speed (10 times faster). Code is available at https://github.com/hhhusiyi-monash/UPDeT

## 1 Introduction

Reinforcement Learning (RL) provides a framework for decision-making problems in an interactive environment, with applications including robotics control (Hester et al. (2010)), video gaming (Mnih et al. (2015)), auto-driving (Bojarski et al. (2016)), person search (Chang et al. (2018)) and vision-language navigation (Zhu et al. (2020)). Cooperative multi-agent reinforcement learning (MARL), a long-standing problem in the RL context, involves organizing multiple agents to achieve a goal, and is thus a key tool used to address many real-world problems, such as mastering multi-player video games (Peng et al. (2017)) and studying population dynamics (Yang et al. (2017)).

A number of methods have been proposed that exploit an action-value function to learn a multi-agent model (Sunehag et al. (2017), Rashid et al. (2018), Du et al. (2019), Mahajan et al. (2019), Hostallero et al. (2019), Zhou et al. (2020), Yang et al. (2020)). However, current methods have poor representation learning ability and fail to exploit the common structure underlying the tasks this is because they tend to treat observation from different entities in the environment as an integral part of the whole. Accordingly, they give tacit support to the assumption that neural networks are able to automatically decouple the observation to find the best mapping between the whole observation and policy. Adopting this approach means that they treat all information from other agents or different parts of the environment in the same way. The most commonly used method involves concatenating

---
*Corresponding author.

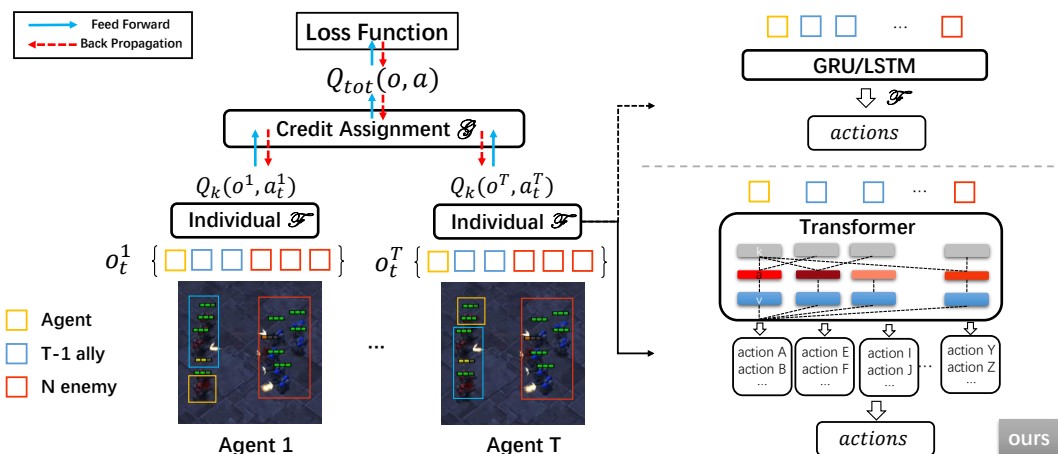

Figure 1: An overview of the MARL framework. Our work replaces the widely used GRU/LSTM-based individual value function with a transformer-based function. Actions are separated into action groups according to observations.

the observations from each entity in to a vector that is used as input (Rashid et al. (2018), Du et al. (2019), Zhou et al. (2020)). In addition, current methods ignore the rich physical meanings behind each action. Multi-agent tasks feature a close relationship between the observation and output. If the model does not decouple the observation from the different agents, individual functions maybe misguided and impede the centralized value function. Worse yet, conventional models require the input and the output dimensions to be fixed (Shao et al. (2018), Wang et al. (2020)), which makes zero-shot transfer impossible. Thus, the application of current methods is limited in real-world applications.

Our solution to these problems is to develop a multi-agent reinforcement learning (MARL) framework with no limitation on input or output dimension. Moreover, this model should be general enough to be applicable to any existing MARL methods. More importantly, the model should be explainable and capable of providing further improvement for both the final performance on single-task scenarios and transfer capability on multi-task scenarios.

Inspired by the self-attention mechanism (Vaswani et al. (2017)), we propose a transformer-based MARL framework, named Universal Policy Decoupling Transformer (**UPDeT**). There are four key advantages of this approach: 1) Once trained, it can be universally deployed; 2) it provide more robust representation with a policy decoupling strategy; 3) it is more explainable; 4) it is general enough to be applied on any MARL model. We further design a transformer-based function to handle various observation sizes by treating individual observations as "observation-entities". We match the related observation-entity with action-groups by separating the action space into several action-groups with reference to the corresponding observation-entity, allowing us to get matched observation-entity — action-group pairs set. We further use a self-attention mechanism to learn the relationship between the matched observation-entity and other observation-entities. Through the use of self-attention map and the embedding of each observation-entity, UPDeT can optimize the policy at an action-group level. We refer to this strategy as **Policy Decoupling**. By combining the transformer and policy decoupling strategies, UPDeT significantly outperforms conventional RNN-based models.

In UPDeT, there is no need to introduce any new parameters for new tasks. We also prove that it is only with decoupled policy and matched observation-entity — action-group pairs that UPDeT can learn a strong representation with high transfer capability. Finally, our proposed UPDeT can be plugged into any existing method with almost no changes to the framework architecture required, while still bringing significant improvements to the final performance, especially in hard and complex multi-agent tasks.

The main contributions of this work are as follows: First, our UPDeT-based MARL framework outperforms RNN-based frameworks by a large margin in terms of final performance on state-of-

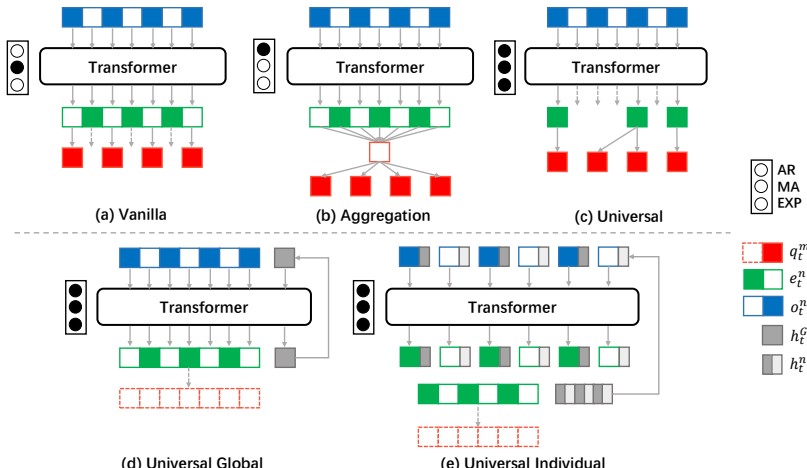

Figure 2: Three variants on different policy decoupling method types (upper part) and two variants on different temporal unit types (bottom). 'AR' , 'MA' and 'EXP' represent Action Restriction, Multi-task at A time and EXPlainable, respectively. $o$, $e$, $q$ and $h$ represents for observation, embedding, q-value and hidden states with $n$ observation entities and $m$ available actions. $G$ represents for the global hidden state and $t$ is the current time step. A black circle indicates that the variant possesses this attribute; moreover, variant (d) is our proposed UPDeT with best performance. Further details on all five variants can be found in Section 3.

the-art centralized functions. Second, our model has strong transfer capability and can handle a number of different tasks at a time. Third, our model accelerates the transfer learning speed (total steps cost) to make it roughly 10 times faster compared to RNN-based models in most scenarios.

## 2 RELATED WORK

Attention mechanisms have become an integral part of models that capture global dependencies. In particular, self-attention (Parikh et al. (2016)) calculates the response at a specific position in a sequence by attending to all positions within this sequence. Vaswani et al. (2017) demonstrated that machine translation models can achieve state-of-the-art results solely by using a self-attention model. Parmar et al. (2018) proposed an Image Transformer model that applies self-attention to image generation. Wang et al. (2018) formalized self-attention as a non-local operation in order to model the spatial-temporal dependencies in video sequences. In spite of this, self-attention mechanisms have not yet been fully explored in multi-agent reinforcement learning.

Another line of research is multi-agent reinforcement learning (MARL). Existing work in MARL focuses primarily on building a centralized function to guide the training of individual value function (Lowe et al. (2017), Sunehag et al. (2017), Rashid et al. (2018), Mahajan et al. (2019), Hostallero et al. (2019), Yang et al. (2020), Zhou et al. (2020)). Few works have opted to form a better individual functions with strong representation and transfer capability. In standard reinforcement learning, this generalization has been fully studied (Taylor & Stone (2009), Ammar et al. (2012), Parisotto et al. (2015), Gupta et al. (2017), Da Silva & Costa (2019)). While multi-agent transfer learning has been proven to be more difficult than the single-agent scenario (Boutsioukis et al. (2011), Shao et al. (2018), Vinyals et al. (2019)). However, the transfer capability of a multi-agent system is of greater significance due to the various number of agents, observations sizes and policy distributions.

To the best of our knowledge, we are the first to develop a multi-agent framework capable of handling multiple task at a time. Moreover, we provide a policy decoupling strategy to further improve the model performance and facilitate the multi-agent transfer learning, which is a significant step towards real world multi-agent applications.

## 3 METHOD

We begin by introducing the notations and basic task settings necessary for our approach. We then describe a transformer-based individual function and policy decoupling strategy under MARL. Finally, we introduce different temporal units and assimilate our Universal Policy Decoupling Transformer (UPDeT) into Dec-POMDP.

### 3.1 NOTATIONS AND TASK SETTINGS

**Multi-agent Reinforcement Learning** A cooperative multi-agent task is a decentralized partially observable Markov decision process (Oliehoek et al. (2016)) with a tuple $G = \langle S, A, U, P, r, Z, O, n, \gamma \rangle$. Let $S$ denote the global state of the environment, while $A$ represents the set of $n$ agents and $U$ is the action space. At each time step $t$, agent $a \in \mathbf{A} \equiv \{1, ..., n\}$ selects an action $u \in U$, forming a joint action $\mathbf{u} \in \mathbf{U} \equiv U^n$, which in turn causes a transition in the environment represented by the state transition function $P(s'|s, \mathbf{u}) : S \times \mathbf{U} \times S \to [0, 1]$. All agents share the same reward function $r(s, \mathbf{u}) : S \times \mathbf{U} \to R$ , while $\gamma \in [0, 1)$ is a discount factor. We consider a partially observable scenario in which each agent makes individual observations $z \in Z$ according to the observation function $O(s, a) : S \times A \to Z$. Each agent has an action-observation history that conditions a stochastic policy $\pi^t$, creating the following joint action value: $Q^\pi(s_t, \mathbf{u}_t) = \mathbb{E}_{s_{t+1:\infty}, \mathbf{u}_{t+1:\infty}} [R_t | s_t, \mathbf{u}_t]$, where $R_t = \sum_{i=0}^\infty \gamma^i r_{t+i}$ is the discounted return.

**Centralized training with decentralized execution** Centralized training with decentralized execution (CTDE) is a commonly used architecture in the MARL context. Each agent is conditioned only on its own action-observation history to make a decision using the learned policy. The centralized value function provides a centralized gradient to update the individual function based on its output. Therefore, a stronger individual value function can benefit the centralized training.

### 3.2 TRANSFORMER-BASED INDIVIDUAL VALUE FUNCTION

In this section, we present a mathematical formulation of our transformer-based model UPDeT. We describe the calculation of the global Q-function with self-attention mechanism. First, the observation $O$ is embedded into a semantic embedding to handle the various observation space. For example, if an agent $a_i$ observes $k$ other entities $\{o_{i,1}, ..., o_{i,k}\}$ at time step $t$, all observation entities are embedded via an embedding layer $E$ as follows:

$$e_i^t = \{E(o_{i,1}^t), ..., E(o_{i,k}^t)\}. \tag{1}$$

Here, $i$ is the index of the agent, $i \in \{1, ..., n\}$. Next, the value functions $\{Q_1, ..., Q_n\}$ for the $n$ agents for each step are estimated as follows:

$$q_i^t = Q_i(h_i^{t-1}, e_i^t, \mathbf{u}_t). \tag{2}$$

We introduce $h_i^{t-1}$, the temporal hidden state at the last time step $t - 1$, since POMDP policy is highly dependent on the historical information. $e_i^t$ denotes the observation embedding, while $u_i^t$ is the candidate action, $u_i^t \in U$. $\theta_i$ is the parameter that defines $Q_i$. Finally, the global Q-function $Q_\pi$ is calculated by all individual value functions, as follows:

$$Q_\pi(s_t, \mathbf{u}_t) = F(q_1^t, .., q_n^t) \tag{3}$$

$F_i$ is the credit assignment function for defined by $\phi_i$ for each agent $a_i$, as utilized in Rashid et al. (2018) and Sunehag et al. (2017). For example, in VDN, $F$ is a sum function that can be expressed as $F(q_1^t, .., q_n^t) = \sum_{i=1}^n q_i^t$.

**Implement Q-function with Self-attention** Vaswani et al. (2017) adopts three matrices, $\mathbf{K}$, $\mathbf{Q}$, $\mathbf{V}$ representing a set of keys, queries and values respectively. The attention is computed as follows:

$$\text{Attention}(\mathbf{Q}, \mathbf{K}, \mathbf{V}) = \text{softmax}(\frac{\mathbf{Q}\mathbf{K}^T}{\sqrt{d_k}})\mathbf{V}, \tag{4}$$

where $d_k$ is a scaling factor equal to the dimension of the key. In our method, we adopt the self-attention to learn the features and relationships from the observation entity embedding and the global temporal information. To learn the independent policy in decentralized multi-agent learning, we

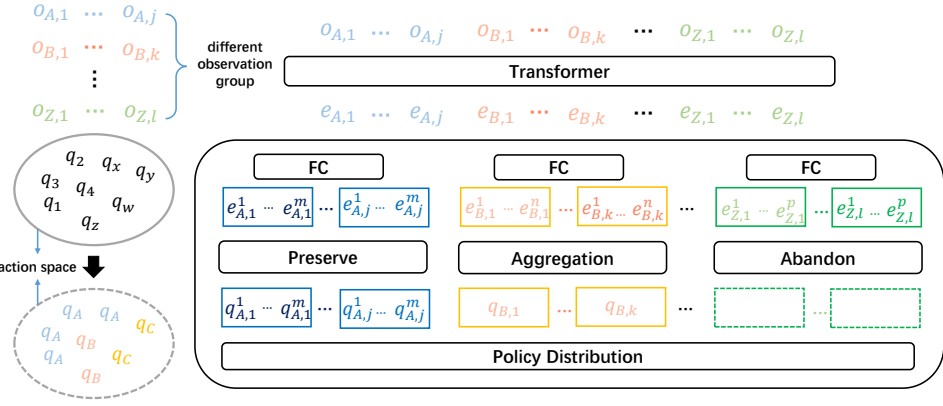

Figure 3: The main pipeline of our proposed UPDeT, where $o, e, q$ represent observation entity, feature embedding and Q-value of each action respectively. Three operations are adopted to avoid introducing new parameters when forming the policy distribution, namely 'preserve', 'aggregation' and 'abandon'. Details can be found in Section 3.3 and a real case can be found in Fig. 7.

define $K_i$, $Q_i$ and $V_i$ as the key, query and value metrics for each agent $a_i$. We further consider the query, key and value for the same matrices $R_i^l = K_i = Q_i = V_i$, where $l \in \{1, ..., L\}$ is the number of layers of the transformer. Thus, we formulate our transformer as follows:

$$R_i^1 = \{h_i^{t-1}, e_i^t\}$$
$$Q_i^l, K_i^l, V_i^l = LF_{Q,K,V}(R_i^l) \tag{5}$$
$$R_i^{l+1} = \text{Attention}(Q_i^l, K_i^l, V_i^l).$$

where $LF$ represents the linear functions used to compute $\mathbf{K}$, $\mathbf{Q}$, $\mathbf{V}$. Finally we project the entity features of the last transformer layer $R_i^L$ to the output space of the value function $Q_i$. We implement the projection using a linear function $P$:

$$Q_i(h_i^{t-1}, e_i^t, u_i) = P(R_i^L, u_i). \tag{6}$$

## 3.3 POLICY DECOUPLING

A single transformer-based individual function with self-attention mechanism is still unable to handle various required policy distribution. A flexible mapping function $P$ in Eq. 6 is needed to deal with the various input and output dimensions and provide strong representation ability. Using the correlation between input and output, we design a strategy called policy decoupling, which is the key part of UPDeT.

The main idea behind the policy decoupling strategy can be summarized into three points:

- Point ①: No restriction on policy dimension. The output dimension of a standard transformer block must be equal to or less than the input dimension. This is unacceptable in some MARL tasks, as the action number can be larger than the entity number.

- Point ②: Ability to handle multiple tasks at a time. This requires a fixed model architecture without new parameters being introduced for new tasks. Unfortunately, if point ① is satisfied, point ② becomes very problematic to achieve. The difficulty lies in how to reconcile points ① and ②.

- Point ③: Make the model more explainable. It would be preferable if we can could replace the conventional RNN-based model with a more explainable policy generation structure.

Following the above three points, we propose three policy decoupling methods, namely Vanilla Transformer, Aggregation Transformer and Universal Policy Decoupling Transformer (UPDeT). The pipelines are illustrated in Fig. 2. The details of the **Vanilla Transformer** and **Aggregation Transformer** are presented in the experiment section and act as our baselines. In this section, we mainly discuss the mechanism of our proposed **UPDeT**.

Tasking the entity features of the last transformer layer outlined in Eq. 5, the main challenge is to build a strong mapping between the features and the policy distribution. UPDeT first matches the input entity with the related output policy part. This correspondence is easy to find in the MARL task, as interactive action between two agents is quite common. Once we match the corresponding entity features and actions, we substantially reduce the burden of model learning representation using the self-attention mechanism. Moreover, considering that there might be more than one interactive actions of the matched entity feature, we separate the action space into several action groups, each of which consists several actions matched with one entity. The pipeline of this process is illustrated in the left part of Fig. 3. In the mapping function, to satisfy point ① and point ②, we adopt two strategies. First, if the action-group of one entity feature contains more than one action, a shared fully connected layer is added to map the output to the action number dimension. Second, if one entity feature has no corresponding action, we abandon it, there is no danger of losing the information carried by this kind of entity feature, as the transformer has aggregated the information necessary to each output. The pipeline of UPDeT can be found in the right part of Fig. 3. With UPDeT, there is no action restriction and no new parameter introduced in new scenarios. A single model can be trained on multiple tasks and deployed universally. In addition, matching the corresponding entity feature and action-group satisfies point ③, as the policy is explainable using an attention heatmap, as we will discuss in Section 4.4.

### 3.4 TEMPORAL UNIT STRUCTURE

Notably, however a transformer-based individual value function with policy decoupling strategy cannot handle a partial observation decision process without trajectory or history information. In Dec-POMDP (Oliehoek et al. (2016)), each agent $a$ chooses an action according to $\pi^a(u^a|\tau^a)$, where $u$ and $\tau$ represents for action and action-observation history respectively. In GRU and LSTM, we adopt a hidden state to hold the information of the action-observation history. However, the combination of a transformer block and a hidden state has not yet been fully studied. In this section, we provide two approaches to handling the hidden state in UPDeT:

1) **Global** temporal unit treats the hidden state as an additional input of the transformer block. The process is formulated in a similar way to Eq. 5 with the relation: $R^1 = \{h_G^{t-1}, e_1^t\}$ and $\{h_G^t, e_L^t\} = R^L$. Here, we ignore the subscript $i$ and instead use $G$ to represent 'global'. The global temporal unit is simple but efficient, and provides us with robust performance in most scenarios.

2) **Individual** temporal unit treats the hidden state as the inner part of each entity. In other words, each input maintains its own hidden state, while each output projects a new hidden state for the next time step. The individual temporal unit uses a more precise approach to controlling history information as it splits the global hidden state into individual parts. We use $j$ to represent the number of entities. The relation of input and output is formulated as $R^1 = \{h_1^{t-1}...h_j^{t-1}, e_1^t\}$ and $\{h_1^t...h_j^t, e_L^t\} = R^L$. However, this method introduces the additional burden of learning the hidden state independently for each entity. In experiment Section 4.1.2, we test both variants and discuss them further.

### 3.5 OPTIMIZATION

We use the standard squared $TD\ error$ in DQNs (Mnih et al. (2015)) to optimize our entire framework as follows:

$$L(\theta) = \sum_{i=1}^{b} \left[ \left( y_i^{DQN} - Q(s, u; \theta) \right)^2 \right] \tag{7}$$

Here, $b$ represents the batch size. In partially observable settings, agents can benefit from conditioning on action-observation history. Hausknecht & Stone (2015) propose Deep Recurrent Q-networks (DRQN) for this sequential decision process. For our part, we replace the widely used GRU (Chung et al. (2014))/LSTM (Hochreiter & Schmidhuber (1997)) unit in DRQN with a transformer-based temporal unit and then train the whole model.

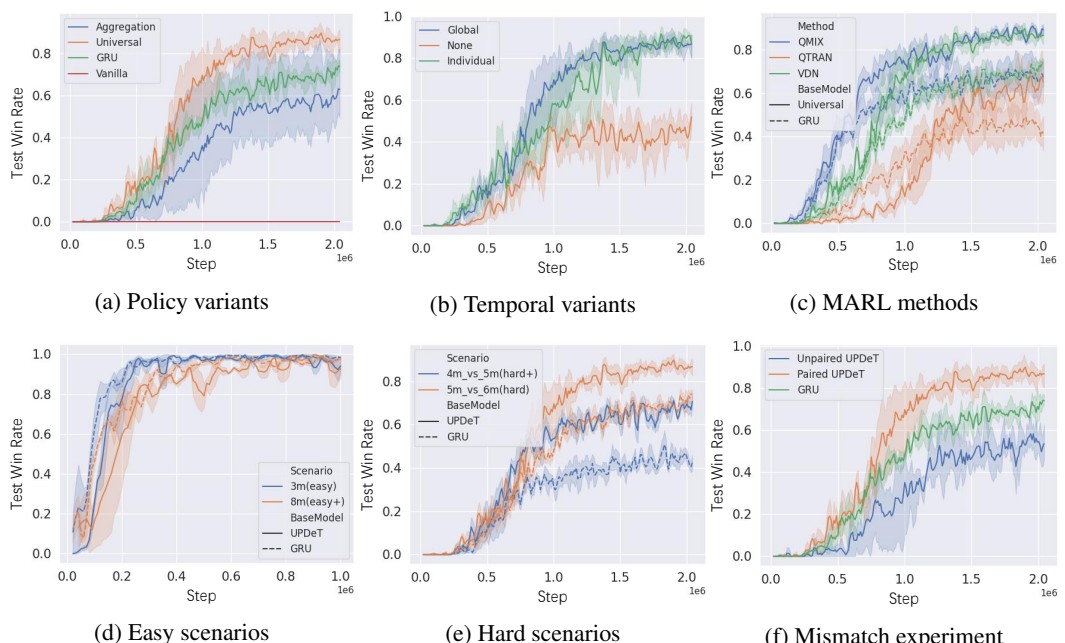

Figure 4: Experimental results with different task settings. Details can be found in Section 4.1.2.

# 4 STARCRAFT II EXPERIMENT

In this section, we evaluate UPDeT and its variants with different policy decoupling methods in the context of challenging micromanagement games in StarCraft II. We compare UPDeT with the RNN-based model on a single scenario and test the transfer capability on multiple-scenario transfer tasks. The experimental results show that UPDeT achieves significant improvement compared to the RNN-based model.

## 4.1 SINGLE SCENARIO

In the single scenario experiments, we evaluate the model performance on different scenarios from SMAC (Samvelyan et al. (2019)). Specifically, the scenarios considered are as follows: 3 Marines vs 3 Marines (3m, Easy), 8 Marines vs 8 Marines (8m, Easy), 4 Marines vs 5 Marines (4m_vs_5m, Hard+) and 5 Marines vs 6 Marines (5m_vs_6m, Hard). In all these games, only the units from player's side are treated as agents. Dead enemy units will be masked out from the action space to ensure that the executed action is valid. More detailed settings can be acquired from the SMAC environment (Samvelyan et al. (2019)).

### 4.1.1 METHODS AND TRAINING DETAILS

The MARL methods for evaluation include VDN (Sunehag et al. (2017)), QMIX (Rashid et al. (2018)) and QTRAN (Hostallero et al. (2019)). All three SOTA methods' original implementation can be found at https://github.com/oxwhirl/pymarl. These methods were selected due to their robust performance across different multi-agent tasks. Other methods, including COMA (Foerster et al. (2017)) and IQL (Tan (1993)) do not perform stable across in all tasks, as have been proved in several recent works (Rashid et al. (2018), Mahajan et al. (2019), Zhou et al. (2020)). Therefore, we combined UPDeT with VDN, QMIX and QTRAN to prove that our model can improve the model performance significantly compared to the GRU-based model.

### 4.1.2 RESULT

The model performance result with different policy decoupling methods can be found in Fig. 4a. **Vanilla Transformer** is our baseline for all transformer-based models. This transformer only satisfies point ②. Each output embedding can either be projected to an action or abandoned. The vanilla

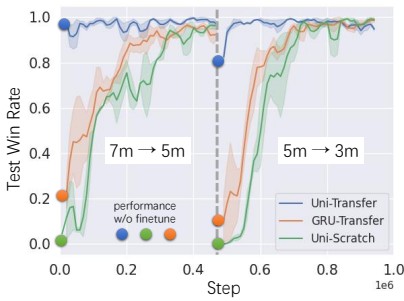 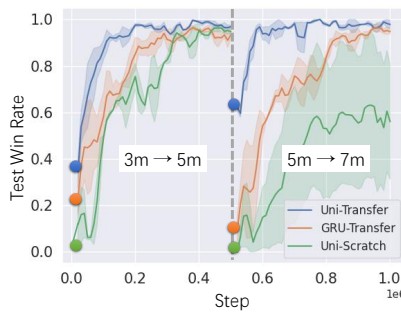

(a) Transfer from 7 marines to 3 marines   (b) Transfer from 3 marines to 7 marines

Figure 5: Experimental results on transfer learning with UPDeT (Uni-Transfer) and GRU unit (GRU-Transfer), along with UPDeT training from scratch (Uni-Scratch). At time step 0 and 500k, we load the model from the source scenario and finetune on the target scenarios. The circular points indicate the model performance on new scenarios without finetuning.

transformer fails to beat the enemies in the experiment. **Aggregation Transformer** is a variant of vanilla transformer, the embedding of which are aggregated into a global embedding and then projected to a policy distribution. This transformer only satisfies the point ①. The performance of the aggregation transformer is worse than that of the GRU-based model. The result proves that it is only with a policy decoupling strategy that the transformer-based model can outperform the conventional RNN-based model. Next, we adopt UPDeT to find the best temporal unit architecture in Fig. 4b. The result shows that without a hidden state, the performance is significantly decreased. The temporal unit with global hidden state is more efficient in terms of convergence speed than the individual hidden state. However, the final performances are almost the same. To test the generalization of our model, we combine the UPDeT with VDN / QMIX / QTRAN respectively and compare the final performance with RNN-based methods in Fig. 4c. We evaluate the model performance on 5m_vs_6m (Hard) scenarios. Combined with UPDeT, all three MARL methods obtain significant improvement by large margins compared to the GRU-based model. The result proves that our model can be injected into any existing stat- of-the-art MARL method to yield better performance. Further more, we combine UPDeT with VDN and evaluate the model performance on different scenarios from Easy to Hard+ in Fig. 4d and Fig. 4e. The results show that the UPDeT performs stably on easy scenarios and significantly outperforms the GRU-based model on hard scenarios, in the 4m_vs_5m(Hard+) scenario, the performance improvement achieved by UPDeT relative to the GRU-based model is of the magnitude of around 80%. Finally, we conduct an ablation study on UPDeT with paired and unpaired observation-entity—action-group, the result of which are presented in Fig. 4f. We disrupt the original correspondence between 'attack' action and enemy unit. The final performance is heavily decreased compared to the original model, and is even worse than the GRU-based model. We accordingly conclude that only with policy decoupling and a paired observation-entity—action-group strategy can UPDeT learn a strong policy.

## 4.2 MULTIPLE SCENARIOS

In this section, we discuss the transfer capability of UPDeT compared to the RNN-based model. We evaluate the model performance in a curriculum style. First, the model is trained one the 3m (3 Marines vs 3 Marines) scenario. We then used the pretrained 3m model to continually train on the 5m (5 Marines vs 5 Marines) and 7m (7 Marines vs 7 Marines) scenarios. We also conduct a experiment in reverse from 7m to 3m. During transfer learning, the model architecture of UPDeT remains fixed. Considering that the RNN-based model cannot handle various input and output dimensions, we modify the architecture of the source RNN model when training on the target scenario. We preserve the parameters of the GRU cell and initialize the fully connected layer with proper input and output dimensions to fit the new scenario. The final results can be seen in Fig. 5a and Fig. 5b. Our proposed UPDeT achieves significantly better results than the GRU-based model. Statistically, UPDeT's total timestep cost to converge is at least 10 times less than the GRU-based model and 100 times less than training from scratch. Moreover, the model demonstrates a strong generalization ability without finetuning, indicating that UPDeT learns a robust policy with meta-level skill.

### 4.3 EXTENSIVE EXPERIMENT ON LARGE-SCALE MAS

To evaluate the model performance in large-scale scenarios, we test our proposed UPDeT on the 10m_vs_11m and 20m_vs_21m scenarios from SMAC and a 64_vs_64 battle game in the MAgent Environment (Zheng et al. (2017)). The final results can be found in Appendix E.

### 4.4 ATTENTION BASED STRATEGY: AN ANALYSIS

The significant performance improvement achieved by UPDeT on the SMAC multi-agent challenge can be credited to the self-attention mechanism brought by both transformer blocks and the policy decoupling strategy in UPDeT. In this section, we mainly discuss how the attention mechanism assists in learning a much more robust and explainable strategy. Here, we use the 3 Marines vs 3 Marines game (therefore, the size of the raw attention matrix is 6x6) as an example to demonstrate how the attention mechanism works. As mentioned in the caption of Fig. 6, we simplify the raw complete attention matrix to a grouped attention matrix. Fig. 6b presents the three different stages in one episode including *Game Start*, *Attack* and *Survive*, with their corresponding attention matrix and strategies. In the *Game Start* stage, the highest attention is in line 1 col 3 of the matrix, indicating that the agent pays more attention to its allies than its enemies. This phenomenon can be interpreted as follows: in the startup stage of one game, all the allies are spawned at the left side of the map and are encouraged to find and attack the enemies on the right side In the *Attack* stage, the highest attention is in line 2 col 2 of the matrix, which indicates that the enemy is now in the agent's attack range; therefore, the agent will attack the enemy to get more rewards. Surprisingly, the agent chooses to attack the enemy with the lowest health value. This indicates that a long term plan can be learned based on the attention mechanism, since killing the weakest enemy first can decrease the punishment from the future enemy attacks. In the *Survive* stage, the agent's health value is low, meaning that it needs to avoid being attacked. The highest attention is located in line 1 col 1, which clearly shows that the most important thing under the current circumstances is to stay alive. For as long as the agent is alive, there is still a chance for it to return to the front line and get more reward while enemies are attacking the allies instead of the agent itself.

In conclusion, the self-attention mechanism and policy decoupling strategy of UPDeT provides a strong and clear relation between attention weights and final strategies. This relation can help us better understand the policy generation based on the distribution of attention among different entities. An interesting idea presents itself here: namely, if we can find a strong mapping between attention matrix and final policy, the character of the agent could be modified in an unsupervised manner.

## 5 CONCLUSION

In this paper, we propose UPDeT, a universal policy decoupling transformer model that extends MARL to a much broader scenario. UPDeT is general enough to be plugged into any existing MARL method. Moreover, our experimental results show that, when combined with UPDeT, existing state-of-the-art MARL methods can achieve further significant improvements with the same training pipeline. On transfer learning tasks, our model is 100 times faster than training from scratch and 10 times faster than training using the RNN-based model. In the future, we aim to develop a centralized function based on UPDeT and apply the self-attention mechanism to the entire pipeline of MARL framework to yield further improvement.

ACKNOWLEDGMENTS

This work was supported in part by the National Natural Science Foundation of China (NSFC) under Grant No.U19A2073 and in part by the National Natural Science Foundation of China (NSFC) under Grant No.61976233 and No.61906109 and Australian Research Council Discovery Early Career Researcher Award (DE190100626), and Funding of "Leading Innovation Team of the Zhejiang Province" (2018R01017).

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

## A    DETAILS OF SMAC ENVIRONMENT

The action space contains four movement directions, k attack actions (where k is the fixed maximum number of the enemy units in a map), stop and none-operation. At each time step, the agents receive a joint team reward, which is defined by the total damage incurred by the agents and the total damage from the enemy side. Each agent is described by several attributes, including health point $HP$, weapon cool down (CD), unit type, last action and the relative distance of the observed units. The enemy units are described in the same way except that CD is excluded. The partial observation of an agent comprises the attributes of the units, including both the agents and the enemy units, that exist within its view range, which is a circle with a specific radius.

## B    DETAILS OF MODEL

The transformer block in all different experiments consists of 3 heads and 2 layer transformer blocks. The other important training hyper parameters are as follows:

| List of Hyper Parameters | |
|---|---|
| Name | Value |
| batch size | 32 |
| test interval | 2000 |
| gamma | 0.99 |
| buffer size | 5000 |
| token dimension (UPDeT) | 32 |
| channel dimension (UPDeT) | 32 |
| epsilon start | 1.0 |
| epsilon end | 0.05 |
| rnn hidden dimension | 64 |
| target net update interval | 200 |
| mixing embeddding dimension (QMIX) | 32 |
| hypernet layers (QMIX) | 2 |
| hypernet embedding (QMIX) | 64 |
| mixing embeddding dimension (QTRAN) | 32 |
| opt loss (QTRAN) | 1 |
| nopt min loss (QTRAN) | 0.1 |

## C    SOTA MARL VALUE-BASED FRAMEWORK

The three SOTA method can be briefly summarized as follows:

- VDN (Sunehag et al. (2017)): this method learns an individual Q-value function and represents $Q_{tot}$ as a sum of individual Q-value functions that condition only on individual observations and actions.

- QMIX (Rashid et al. (2018)): this method learns a decentralized Q-function for each agent, with the assumption that the centralized Q-value increases monotonically with the individual Q-values.

- QTRAN (Hostallero et al. (2019)): this method formulates multi-agent learning as an optimization problem with linear constraints and relaxes it with L2 penalties for tractability.

## D    UPDeT ON SMAC: A REAL CASE

We take the 3 Marines vs 3 Marines challenge from SMAC with UPDeT as an example; more details can be found in Fig. 7. The observation are separated into 3 groups: main agent, two other ally agents and three enemies. The policy output includes basic action corresponding to the main agent's observation and attack actions, one for each enemy observation. The hidden state is added after the embedding layer. The output of other agents is abandoned as there is no corresponding

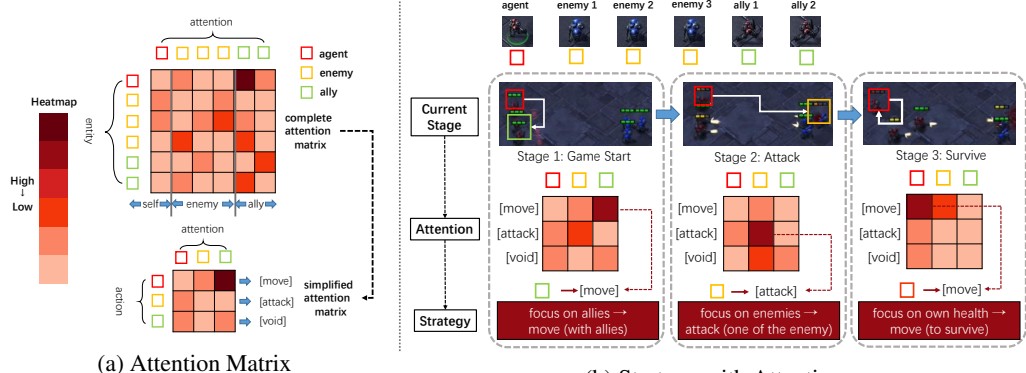

(a) Attention Matrix          (b) Strategy with Attention

Figure 6: An analysis of the attention based strategy of UPDeT. Part (a) visualizes a typical attention matrix. Part (b) utilizes the simplified attention matrix to describe the relationship between attention and final strategy. Further discussion can be found in Section 4.4.

action. Once an agent or enemy has died, we mask corresponding unavailable action in the action select stage to ensure only the available actions are selected.

# E   RESULTS OF EXTENSIVE EXPERIMENT ON LARGE SCALE

We further test the robustness of UPDeT in a large-scale multi-agent system. To do so, we enlarge the game size in SMAC (Samvelyan et al. (2019)) to incorporate more agents and enemies on the battle field. We use a 10 Marines vs 11 Marines game and a 20 Marines vs 21 Marines game to compare the performance between the UPDeT and GRU-based approaches. In the 20 Marines vs 21 Marines game, to accelerate the training and satisfy the hardware limitations, we decrease the batch size of both the GRU baseline and UPDeT from 32 to 24 in the training stage. The final results can be found in Fig. 8a. The improvement is still significant in terms of both sample efficiency and final performance. Moreover, it is also worth mentioning that the model size of UPDeT stays fixed, while the GRU-based model becomes larger in large-scale scenarios. In the 20 Marines vs 21 Marines game, the model size of GRU is almost double that of UPDeT. This indicates that UPDeT is able to ensure the lightness of the model while still maintaining good performance.

We also test the model performance in the MAgent Environment (Zheng et al. (2017)). The settings of MAgent are quite different from those of SMAC. First, the observation size and number of available actions are not related to the number of agents. Second, the 64_vs_64 battle game we tested is a two-player zero-sum game which is another hot research area that combines both MARL and GT (Game Theory), the most successful attempt in this area involves adopting a mean-field approximation of GT in MARL to accelerate the self-play training (Yang et al. (2018)). Third, as for the model architecture, there is no need to use a recurrent network like GRU in MAgent and the

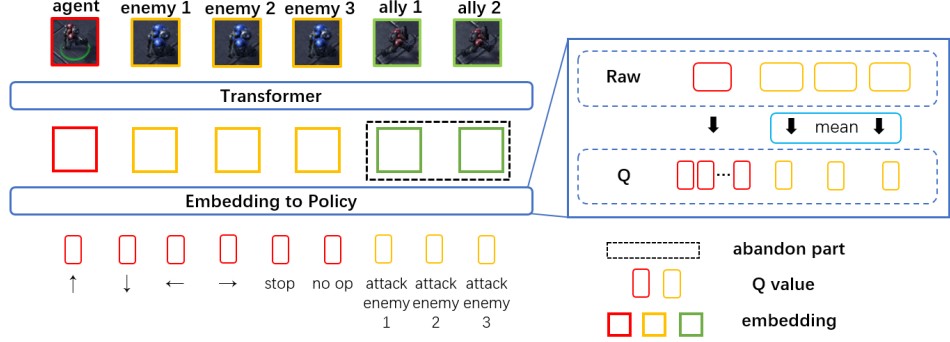

Figure 7: Real case on 3 Marines vs 3 Marines Challenge from SMAC.

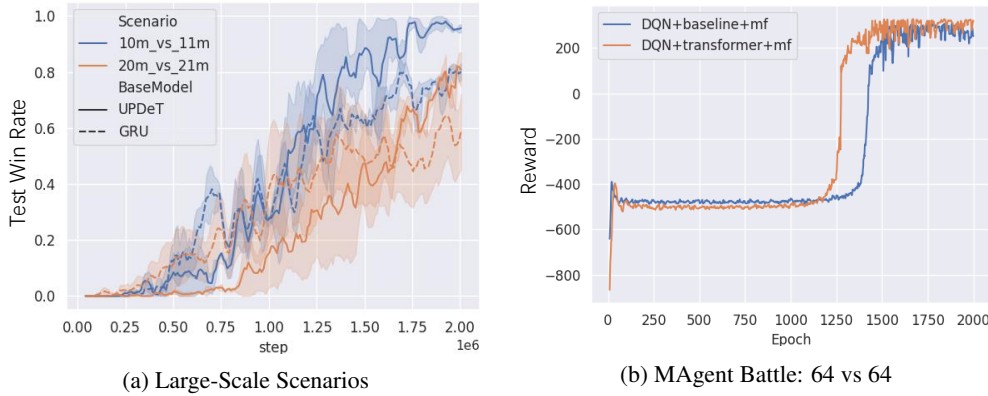

(a) Large-Scale Scenarios

(b) MAgent Battle: 64 vs 64

Figure 8: Experimental results on the large-scale MAS, including SMAC and MAgent.

large observation size requires the use of a CNN from embedding. However, ny treating UPDeT as a pure encoder without recurrent architecture, we can still conduct experiments on MAgent; the final results of these can be found in Fig. 8b. As the result show, UPDeT performs better than the DQN baseline, although this improvement is not as significant as it in SMAC.

