# OpenReview forum: "UPDeT: Universal Multi-agent RL via Policy Decoupling with Transformers"
_ICLR.cc/2021/Conference — ICLR 2021 Spotlight_

### Official Review · AnonReviewer1 · 2020-10-27
**A promising transformer-based MARL architecture with convincing results on SMAC. Clarity and readability need to be improved.**

**Rating:** 7
**Confidence:** 4

**Review:**


### Summary and claims of the paper

The authors propose several Transformer-based architectures that can be used in combination with a MARL (multi-agent reinforcement learning) algorithm to tackle multi-agent environments. They identify a particularly suitable architecture they call UPDeT which they claim combines the following advantages:
 1) Can be used in the multi-task setting, where the number of entities and therefore the inputs changes from one task to the next.
 2) Can handle a varying number of actions per entity (e.g. the controlled units could have N actions and each enemy unit corresponds to 1 action (which one to target), allied units correspond to no actions).
 3) Is more explainable because the attention weights of the transformer can be inspected to see which information a given output depends on.
 4) Outperforms the standard RNN approach on specific tasks.
 5) Contrary to the standard RNN approach, the architecture is able to zero-shot generalize to tasks with different numbers of entities and learns significantly faster than the RNN architecture when fine-tuning on a new task. (This is in my opinion the most important and interesting claim).

I have collected these claims from different parts of the paper and had to add some of my own interpretation to disambiguate what was meant. Please correct me if I misunderstood any of the claims. It would be nice if all of these were stated clearly in one place in the paper.

Points 1-3 can be argued based on the design of the architecture alone, while points 4) and 5) are supported by experiments using the SMAC (StarCraft Multi-Agent Challenge) environment.

A summary of UPDeT from my understanding:
In MARL a group of independently acting agents needs to be trained to maximize a reward. Each agent has a separate set of Q values for each action that it can issue at a given time. In UPDeT, the Q values for one agent are computed by applying a transformer to all of the entities that the given agent can see. The resulting transformer output is then mapped into Q values depending on the positions in the transformer output. E.g. in the case of SMAC the position of the currently controlled unit corresponds to many actions, like moving left/right/up/down. The position of enemy units corresponds to 1 action each (targeting that unit) and the position of allied units (also controlled by the network, but not by this application of the transformer) corresponds to 0 actions. The mapping from transformer output to N actions is handled through a linear layer that is shared between entity groups (e.g. outputs for all enemy units get transformed with the same linear layer). A recurrent state through time can be added either globally for the whole transformer, or added to the individual units.

### Discussion of prior work

The section on prior work is split between references to transformer and MARL literature.
From what I can tell it is complete enough to position the paper sufficiently in the literature.
The one thing that I think is missing is a reference to the AlphaStar paper (https://www.nature.com/articles/s41586-019-1724-z) since this is an example of an agent that uses a transformer and is also applied to StarCraft.
In contrast to this work it doesn't use a MARL algorithm to control multiple units.

### Are the claims supported?

Claims 1) and 2) are supported directly by the design of the architecture.
I struggled to understand the details of the architecture for some time. In my opinion this part of the paper could really benefit from a clearer presentation of the architecture (more details and some suggestions for improvement are in the next section).

I'm somewhat unsatisfied with how claim 3) is handled in the paper, because the authors don't include any demonstration of explainability (e.g. a particular instance where the attention weights make it clear how a certain decision is made). Simply claiming that the architecture is explainable because a transformer is used is not enough, because in my experience there can be plenty of instances where we can't interpret how a given decision was made through the attention weights. It would be good if the authors would either give an interpretable example, or soften their claims of explainability and remove it from the main set of claims.

Claims 4) and 5) are handled pretty well in my opinion.
In my experience the SMAC challenges are decently complex and allow meaningful differences in performance between algorithms to be measured.
The results in figure 4 support the design decisions in the architecture and show that the proposed architecture outperforms the standard RNN approach.
The only thing I'm unsure about here is the "vanilla transformer" architecture, which doesn't seem to be working at all. Is it possible that the transformer output is not conditioned on the different action types? E.g. this could be achieved by adding a positional encoding to the transformer corresponding to the different actions that can be issued. In any case such an architecture doesn't make a lot of sense since the actions are not related to the entities processed by the transformer. Maybe the aggregation transformer should be the "vanilla" version and the vanilla version can just be removed to make the paper simpler?

Claim 5) is supported by a separate set of experiments where the agents are trained with a given number of units, and then switched to a task with a different number of units. UPDeT still performs reasonably well after the switch (zero-shot generalization) and quickly recovers to full performance during fine-tuning. I found this result genuinely interesting and I like how it is presented in the paper, e.g. by including the from-scratch curves as well.
As pointed out in the paper, the GRU baseline had to have some of its weights reset when switching from one task to another due to the changed action space. I wonder what would happen if a similar technique to UPDeT was used on the GRU, e.g. by emitting actions from different action heads depending on the identity of a unit as the GRU is unrolled (ideally it should be bidirectional). That way the resetting could be avoided and a fairer comparison between RNN and transformer could be made. This is not a crucial addition to the paper, but could be interesting.

### Presentation and clarity

I think the presentation and clarity of the paper should be improved significantly.
There are currently many typos, grammatical mistakes and missing words that made it hard for me to understand the paper.
The sections 3.2, 3.3 and 3.4 in particular are really crucial for understanding the paper and have some mistakes that confused me significantly.
For example in the bottom of formula (5) the attention is applied to three identical variables $R_i^l$ (this is done in other places in the paper as well, I'm not sure how to interpret it).
In Figure 3 the embedding outputs are labeled $e_{B,1}\cdots o_{B,k}$, but I think this should be $e_{B,1}\cdots e_{B,k}$.
I found Figures 2 and 3 very difficult to understand, maybe because they show a very general version of the idea rather than the specific one that was tested by the authors.
The most confusing part of Figure 2 for me was the alternate white/colored filling scheme on the various tensors. What does it mean for a position to be colorless or colored here?
I found Figure 6 in the appendix extremely useful for understanding the architecture. Maybe this should be part of the main text?
I can appreciate that the authors want to present a general view of their idea (a "framework" rather than a specific architecture), but personally I would have understood the paper faster if it would focus on the specific case used for SMAC.
There are some references to "training speed" in the paper. It took me some time to figure out that this refers to the number of update steps rather than the actual speed of training the network. It would be good to clarify that.

Overall I think the impact of the paper could really benefit from improving presentation and clarity.
I have not taken this aspect into account in my rating.

### Conclusion

Pros:
 - Benefit of using a transformer in a MARL setup is demonstrated in a non-trivial environment.
 - The proposed architecture goes beyond "obvious" approaches like the pooling the output of the transformer and the benefit of this is demonstrated.

Cons:
- The paper is currently difficult to understand in several important places.
- I think the use of transformers as presented here is only valid for environments with structured observations, e.g. where actions map nicely to specific entities in the observation.

Overall I think the paper presents a promising approach to building MARL architectures. The approach makes sense and the experiments are insightful, but the presentation of the paper needs to be improved.

*Edits after author comments and revision:*
The authors have greatly improved the readability of the paper. I also appreciate the addition of section 4.4. which seems like a reasonable attempt at supporting the increased interpretability of the architecture. I feel that the paper has improved, but it wasn't quite enough for me to raise the rating from 7 to 8, so I'm leaving it at "Good paper".

---

> ### Author Response · Authors · 2020-11-17
> **Response to AnonReviewer1:**
>
> Thanks for your explicit and helpful comments on almost every aspect of our paper. Here, we address your 3 main concerns.
> 1. Point 3 of 5 claims is not satisfied and well explained.
> We admit that the claim point 3 is not handled very well in our paper. Therefore, we have extended the paper to 9 pages and discussed the attention based policy generation process in Section 4.4, including a complete analysis on 3 marines vs 3 marines demo game using UPDeT as backbone.
> 2. Is “vanilla transformer” necessary to the paper ?
> We agree with you that aggregation transformer can serve as a better baseline compared to vanilla transformer. The only weakness of aggregation transformer is that it can not deal with multi-task setting. Once the embeddings are aggregated, we can not recover the specific entity number. However, we admit that the vanilla transformer needs to be revised for a stronger baseline. We agree with you that this could be achieved by adding a positional encoding to the transformer corresponding to the different actions that can be issued. We will make further improvement on that. Thanks for your constructive advice.
> 3. Bi-directional GRU may be another choice.
> Thank you for this insightful comment. We believe a bi-directional GRU can be a strong competitor against UPDeT as both two architectures perform well in many sequence to sequence tasks. While one limitation of Bi-GRU is obvious: it requires more time to inference as the GRU is serial while the Transformer is parallel.  We are now working on this and will provide a Bi-GRU based “UPDeG” in the future. Many thanks for your suggestion on this point.
>
> Furthermore, we correct the typos, grammatical mistakes and missing words as much as we can and revise the confusing formula (5)(7)(8). We have added AlphaStar paper to the paper reference. It is true that we want to present a more general framework rather than a specific architecture which makes Fig 2 & 3 a bit hard to understand. In figure 2, we have adopted the alternate white/colored filling scheme to indicate that observations are from different entities and should be treated separately. In figure 3, we have corrected the wrong typos. To help better understand the model architecture, we have added a figure reference for figure 7 (a real case of UPDeT) in the figure 3 captions. Besides, we have clarified that the training speed we mentioned in this paper refers to the total steps cost from step zero to convergence.

---

### Official Review · AnonReviewer2 · 2020-10-28
**This paper studies the interesting problem of universal multi-agent reinforcement learning for multiple tasks, utilizing a new transformer-based model approach. It has been demonstrated that the new approach proposed in this paper has significant advantages over the existing state-of-art methods in terms of final performance, training time, as well as transfer capability for multiple tasks.**

**Rating:** 9
**Confidence:** 4

**Review:**

In my opinion, this paper is a very good paper: novel in the approach, high impacts in both theoretic sense and practical sense, and well-written. The problem of universal multi-agent reinforcement learning for multiple tasks is very interesting and challenging, and the methodology proposed in this paper is inspiring and the demonstrated experimental results are very impressive.

Main contributions of this paper are as follows, which are very significant and impressive:
 [1] The proposed UPDeT-based MARL framework outperforms RNN-based frameworks on state-of-the-art centralized functions by a large margin in terms of final performance.
[2] The proposed model has strong transfer capability and can handle a number of different task at a time.
[3] The proposed model accelerates the transfer learning speed so that it is about 10 times faster compared to RNN-based models in most scenarios.

The paper presentation is in good quality, the concepts and the methodologies were explained clearly. The provided experiments section is also convincing and somewhat extensive, described and presented very clearly, which supports the main claimed contribution very well.

The authors also point out some promising future research direction on top of this work, which is also helpful.

Overall, I strongly support this paper to be accepted and published in ICLR. The experiment results are impressive, and the proposed methodologies are inspiring and novel. I believe that many people in the research community would find it valuable/inspiring and benefit from this paper.

---

> ### Author Response · Authors · 2020-11-17
> **Response to AnonReviewer2:**
>
> Thank you for your recognition of our work. Upon acceptance, we will release our code to benefit the research community and expand the coverage area of our paper to make it more practical and easy to understand.

---

### Official Review · AnonReviewer4 · 2020-10-30
**A novel work, but lack of strong experiments**

**Rating:** 6
**Confidence:** 4

**Review:**

1. In this paper the authors proposed a transferrable framework for multi-agent RL, which enables the learned policies easily generalize to more challenging scenarios. This seems to be a good contribution to the community of multi-agent RL. It bears a potential to handle large-scale tasks with only limited training data, while also demonstrates more explanable policies.
2. However, the experiments seem to be insufficient. The authors only investigate scenarios for 3 vs. 3, 5 vs. 7, which are still the easiest cases in the StarCraft II combat tasks. I suggest the authors to try more on 20 vs. 30 StarCraft combat task or more challenging scenarios, or the hundres or thousands levels of multi-agent tasks like that provided by MAgent environment[1]. And a comprehensive comparison with a similar work [2] following the curriculum learning pipeline is also worth a trial. This will make this work a strong one.
3. A more profound analysis is needed for the experiment part. Besides the performance gains, insightful understanding of how the designed model works is also necessary.

[1] Yaodong Yang, Rui Luo, Minne Li, Ming Zhou, Weinan Zhang, Jun Wang. Mean Field Multi-Agent Reinforcement Learning. ICML 2018.
[2] Kun Shao, Yuanheng Zhu, Dongbin Zhao. StarCraft Micromanagement With Reinforcement Learning and Curriculum Transfer Learning. IEEE Transactions on Emerging Topics in Computational Intelligence, 2018.

---

> ### Author Response · Authors · 2020-11-17
> **Response to AnonReviewer4:**
>
> Thanks for your comments. Below we will mainly address your concerns on point 2 and point 3.
>
> Point 2 Comment 1: Only easiest scenarios like 3 vs 3 5 vs 5 are tested.
> Reply 1: The reviewer might have overlooked that we have investigated the model performance on both easy (3v3 8v8) and hard/hard+ (4v5 5v6) scenarios. The related experiment results can be found in Fig 4 (d)(e). In hard+ scenarios, RNN based architecture can only get less than 50% winning rate while UPDeT gets significant improvement on winning rate performance. Therefore, the level of ‘easy’ or ‘hard’ does not depend on agent number but the final winning rate. However, we fully understand that large-scale scenarios are also important to model robustness. We therefore provide 10m_vs_11m and 20m_vs_21m experiment results from SMAC in the Fig.8 (a) at Appendix E. We are not able to do very large scale (>30) experiments on SMAC due to rebuttal time and resource limitations.
>
> Point 2 Comment 2: MAgent environment is another benchmark on MARL.
> Reply 2: MAgent by GeekAI is another good MARL environment which has been employed in several papers. Though the setting of MAgent is different from SMAC, we have added additional experiments in Section 4.3 and the experiment results with discussion in section E of Appendix. We have also included the recommended references [1][2] in the updated version.
>
>
> Point 2 Comment 3: Comprehensive comparison with a similar work [2] following the curriculum learning pipeline is also worth a trial.
> Reply 3: Thanks for pointing this out. If there is a work similar to ours, [2] is the earliest one. And lately, there is another related work [3] which also adopts a curriculum style to transfer the knowledge from few agent tasks to more agent tasks. However there are two significant weaknesses of these approaches compared to UPDeT: First, UPDeT has no need to be trained in ‘curriculum style’ like [2] and [3] did. As mentioned by AnonReviewer 1, our proposed model shows great zero-shot generalization ability with no restrictions on ‘few’ to ‘more’ or ‘easy’ to ‘hard’ training pipeline. Second, both [2][3] are methods based on a fixed action number which means we need to re-initialize the output layer of the source model to fit the new task under SMAC. Therefore, the performance is far lower than UPDeT and nearly has no zero-shot generalization ability. Corresponding proof of this claim can be found in Fig.5 in [3] and Fig.9 in [2]. In their experiments, all the training curves of winning rate start at zero when transferred to new tasks. We compare these two related works in Section 1 to make UPDeT a stronger and more attractive framework to real world application. Thanks for your helpful advice.
>
> [3] Wang W, Yang T, Liu Y, et al. From Few to More: Large-Scale Dynamic Multiagent Curriculum Learning[C]//AAAI. 2020: 7293-7300.
>
> Point 3 Comment: Insightful understanding behind performance gains is needed.
> Reply: AnonReviewer 1 & 4 both point out that we should do additional analysis on experiment results to better understand the UPDeT. However, due to page number limitation, we haven’t explicitly explained the explainity of UPDeT on decision making as claimed in point 3 section 3.3. We have now provided an analysis on self-attention based policy generation process in Section 4.4 at page 8 & 9.  We take 3 marines vs 3 marines as an example for simplicity and show 3 policy generation processes on different stages in one episode.

---

> > ### Comment · AnonReviewer4 · 2020-11-20
> > **The revisions make this submission a stronger one**
> >
> > Based on the supplementary experiments&analysis provided by the authors, I feel the submission now is a stronger one. The proposed framework does bear its superiority and can be inspiring for future research. In the new experiments, the framework also demonstrates its advantage over related works. Hence I decide to promote my rating to 6. I suggest the authors have a full typo-fixing over the paper to avoid readibility concerns.

---

### Author Response · Authors · 2020-11-20
**General Response to Reviewers' Comments**

We thank all reviewers for their thoughtful feedback. We have posted a revision incorporating these feedback.

The first important revised part of the paper is in Section 4.4: “Attention Based Strategy: An Analysis”, which describes the explainity of UPDeT using a real case from SMAC.

The second main revised part is in Appendix E: “Results of Extensive Experiment on Large Scale”, which guarantees the robustness of UPDeT in large-scale scenarios.

Besides, we have invited a native speaker to proofread this paper and corrected all the typos and grammar errors. Formulation [5][7][8] in Section 3 have been rewritten. Other details and discussions can be found in each of our response.

---

### Decision · Program_Chairs · 2021-01-07
**Final Decision**

**Decision:**

Accept (Spotlight)

**Comment:**

Reviewers all agree on acceptance for this paper. The initial issues with clarity seem to have been addressed by the authors.

The paper introduces a new transformer-based architecture for MARL that enables variable input and output sizes, which is used to train the agent in a more general setting and on more diverse tasks for multi-task training. The method also produces more interpretable agents.
The paper shows results on the Starcraft multi-agent challenge (not the full game of Starcraft, but still a recognised and widely used multi-agent benchmark). The method produces solid results both in terms of final training performance and zero-shot generalisation.

Although reviewers are generally supportive of this paper, they mention that the Starcraft challenge used is somewhat simple (only few units used), and that the transformer-based architecture may not be applied to domain which lack the proper structure.